# Unsupervised Anomaly Detection and Segmentation on Dirty Datasets

**Jiahao Guo, Xiaohuo Yu and Lu Wang** *

School of Computer Engineering and Science, Shanghai University, Shanghai 200444, China; g747173965@shu.edu.cn (J.G.); xiaohuoyu@shu.edu.cn (X.Y.)
* Correspondence: luwang@shu.edu.cn

**Abstract:** Industrial quality control is an important task. Most of the existing vision-based unsupervised industrial anomaly detection and segmentation methods require that the training set only consists of normal samples, which is difficult to ensure in practice. This paper proposes an unsupervised framework to solve the industrial anomaly detection and segmentation problem when the training set contains anomaly samples. Our framework uses a model pretrained on ImageNet as a feature extractor to extract patch-level features. After that, we propose a trimming method to estimate a robust Gaussian distribution based on the patch features at each position. Then, with an iterative filtering process, we can iteratively filter out the anomaly samples in the training set and re-estimate the Gaussian distribution at each position. In the prediction phase, the Mahalanobis distance between a patch feature vector and the center of the Gaussian distribution at the corresponding position is used as the anomaly score of this patch. The subsequent anomaly region segmentation is performed based on the patch anomaly score. We tested the proposed method on three datasets containing the anomaly samples and obtained state-of-the-art performance.

**Keywords:** anomaly detection and segmentation; unsupervised method; noise resistance

## 1. Introduction

In the manufacturing production process, quality control is one of the most important aspects. Finding and locating surface defects in industrial products using machine vision is much more efficient than manual quality inspection [1]. Thus, vision-based detection and segmentation of surface defects in industrial products have become a key research topic [2,3]. As a rule, products without surface defects are defined as normal samples, and products with surface defects are defined as anomaly samples. The anomaly detection task is to determine whether a sample contains defects or not. The anomaly segmentation task is to locate the defective region.

Anomaly detection and segmentation have the following challenges. Firstly, anomalies present in various ways, such as irregular, inconsistent, faulty, unnatural, atypical, etc. [4]. Secondly, the camera rarely captures anomaly products in practice leading to data imbalance, i.e., there are many normal samples and a few anomaly samples. Thirdly, anomaly samples are rare entities, hence it is inefficient to manually label them [5].

Due to the abovementioned points, unsupervised methods are used for anomaly detection, and these methods assume that all samples in the training set are normal samples [4]. This assumption is hard to satisfy in practice. The performance of most of the existing unsupervised anomaly detection methods degrades severely in the case of "dirty" datasets, i.e., the training set contains anomaly samples [2,4,5].

We propose a framework to address anomaly detection and segmentation in the case of dirty datasets. We consider samples in the low-density region of the probability model as anomaly samples. Firstly, inspired by the trimming method in robust statistics [6], we fit a Gaussian distribution using patch-level features for each position where the patch is

located and exclude the samples that contain patches in low-density regions, as shown in Figure 1. Secondly, we obtain the final selected normal samples through the iteration module. Lastly, we fit patch-level Gaussian distributions based on the final selected normal samples. In the testing phase, the Mahalanobis distance between each patch of the test sample and the center of the corresponding Gaussian distribution is used as the anomaly score to perform anomaly detection and segmentation. This paper calls this the framework PaIRE (Patch Iterative Robust Estimation) method. The main contributions of our work are as follows:

- Unsupervised anomaly detection methods degrade in performance due to noise. We propose a framework PaIRE to address unsupervised anomaly detection when the training set contains noise;
- A slope sliding window method is designed to reduce the effect of noise for improving the noise resistance of the proposed model;
- An iterative module is proposed to make the model estimate the distribution of normal samples more accurately from the dirty dataset;

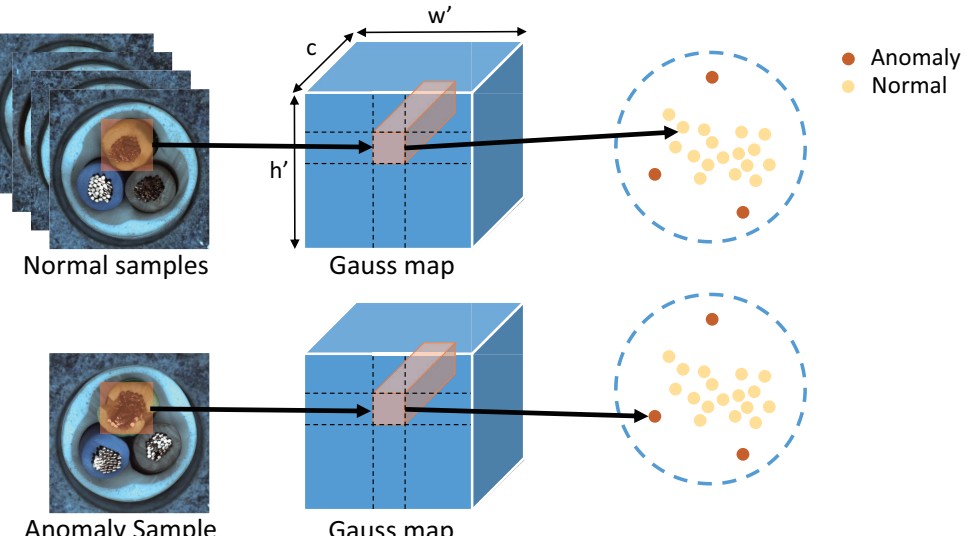

**Figure 1.** The feature vector corresponding to the image patch can be obtained from the feature map extracted from the pre-trained network. In the feature space, the normal patches are in the high-density region and the anomaly patches are in the low-density region.

## 2. Related Works

Unsupervised industrial anomaly detection and segmentation methods can be classified into classification methods, probabilistic methods, and reconstruction methods. In addition to this, a few unsupervised anomaly detection methods are based on dirty datasets. The above methods will be reviewed below.

Most of the unsupervised classification anomaly-detection methods are based on normal samples. OC-SVM [7] treats the coordinate origin as anomaly samples and trains a classifier with the coordinate origin and the normal samples. SVDD [8,9] finds the minimum volume of hypersphere that can enclose most of the samples and considers an anomaly sample outside the hypersphere. Recently, self-supervised classification has been used for anomaly detection. Geometric transformations [10] (rotation, flip, translation, etc.) train the classifier by predicting which transformation the image has performed. Anomaly scores are obtained from the softmax activation of the classifier. RotNet [11] uses rotation for data augmentation. After that, a one-class classifier is obtained by contrast learning [12] for anomaly detection. NSA [13] uses seamless cloning to create anomaly samples and trains a binary classifier to distinguish artificial anomaly samples from normal samples. CutPaste [14] performs anomaly detection by training a classifier with normal samples and

anomaly samples constructed by special data augmentation. Anomaly detection using unsupervised classification methods is sensitive to noise in the training set. When the training set contains noise, both one-class classification methods [7,9] and self-supervised classification methods [10,11,13,14] produce incorrect labels, which seriously affect the detection performance of the model.

In the probabilistic model, anomalies are defined as low probability samples. Gaussian-AD [15] fits a Gaussian distribution at each feature hierarchy of the pretrained network, and the sum of the Mahalanobis distances between the test samples and the center of the corresponding Gaussian distribution at each hierarchy is calculated as the anomaly score. PaDim [16] uses a pre-trained network to extract features, fits a Gaussian distribution at patch-level, and uses Mahalanobis distances as anomaly scores. FYD [17] uses an auxiliary alignment network to align the objects in the images and uses a Patch-level Gaussian classifier for anomaly detection. SPADE [18] also uses a pretraining network to extract features. It uses a memory module to store the features of all training samples, and the Euclidean distance sum of the test sample and the K-NN samples in the memory module as the anomaly score. STPM [19] uses knowledge distillation for anomaly detection: the student network learns the pretrained teacher network based on normal samples. It utilizes the difference between student and teacher extracted features of the anomaly samples for anomaly detection. For parametric probabilistic models, noisy samples affect the estimation of parameters. For nonparametric models, noisy samples are stored in the memory module and mistaken for normal samples, affecting the model performance.

Reconstruction is one of the mainstream anomaly-detection methods based on normal samples. The reconstruction model is trained only on normal samples, which leads to low reconstruction error for normal samples and high reconstruction error for anomaly samples. Generative adversarial networks (GAN) [20–25], autoencoders (AE) [26–30], and variants are the most commonly used reconstruction models. InTra [31] uses a transformer to inpaint the artificial missing image patches. The patches containing anomalies will have high reconstruction errors. The reconstruction error of the test sample is used as the anomaly score. The drawback of the reconstruction method is that when the training set contains anomaly samples, it leads to low reconstruction error for the anomaly samples.

There are few studies on industrial anomaly detection and segmentation in the case of dirty datasets. TrustMAE [32] filters anomaly samples in the autoencoder latent space based on the distance of the sample from the center of the batch samples. Samples with a distance greater than a threshold from the centroid are considered anomaly samples. The disadvantage of TrustMAE is that at the early stage of model training, there are no labels to guide the encoder to extract distinguishing features, which can lead to the possibility that samples close to the center of the batch sample also be anomaly samples. Li et al. [33] use the feature vector of the latent space of autoencoder as a proxy to perform clustering and select clusters with low variance as normal samples. This process is cycled in the training process of the autoencoder. These two methods have a drawback: early in the autoencoder training process, the extracted feature vectors may not satisfy their assumptions. DFR [34] extracts the ImageNet pretrained network features, determines the optimal latent space dimension of AE by PCA, and then uses AE to reconstruct the extracted features. DFR is robust to noise because of the noise-reduction effect of PCA.

## 3. Proposed Method

As shown in Figure 2, our proposed PaIRE (Patch Iterative Robust Estimation) method consists of the following parts: feature extractor, trimming patch-level gaussian distribution (Trimming Module), iterative filtering(Iteration Module), anomaly detection and segmentation (Inference) based on anomaly scores. Each part will be described in the following.

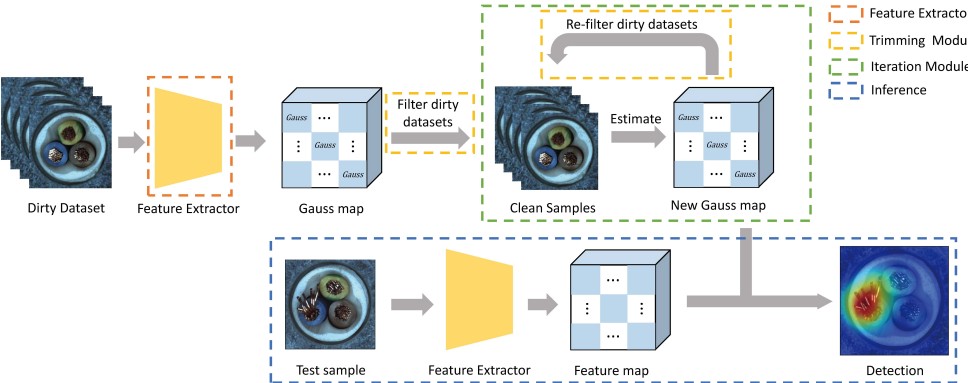

**Figure 2.** The structure of our proposed method PaIRE.

### 3.1. Patch Feature Extraction

For industrial anomaly-detection tasks, a good feature extractor should distinguish the difference between normal and anomaly samples. When the dataset contains anomaly samples without any labels, it is hard to train a good feature extractor. Some recent work uses networks that are pretrained on ImageNet for anomaly detection [15,16,18,34], and these works illustrate the ability of pre-trained networks to extract discriminative features for anomaly detection. We use a network pretrained on ImageNet to generate the patch-embedding vector.

Different blocks produce feature maps containing different hierarchical semantic information in the ResNet-like architecture. Since anomalies may occur at different scales, and to avoid features being too biased towards ImageNet classification, we choose the feature map of blocks 1,2,3 outputs of WideResnet-50 to extract the patched-level features as shown in Figure 3. We use $\phi_q, q \in \{1,2,3\}$ to denote the output of block $q$.

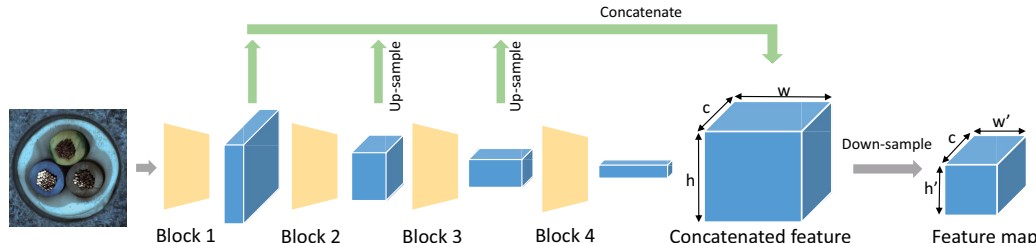

**Figure 3.** Extracting feature map from WideResnet-50 pretrained on ImageNet.

To fuse the information from different hierarchies and align feature maps spatially, we use bilinear interpolation to resize $\phi_2$ and $\phi_3$ to the same resolution on $h$ and $w$ as $\phi_1$, as shown in Equation (1). Here, $\phi_q \in \mathbb{R}^{c \times h \times w}$ is a three-dimensional tensor of channel $c$, height $h$, width $w$.

$$\hat{\phi}_q = resize(\phi_q), q \in \{2,3\} \tag{1}$$

Then we concatenate $\phi_1, \phi_2, \phi_3$ on the channel, as shown in Equation (2).

$$\hat{\phi}_{\{1:3\}} = cat(\phi_1, \hat{\phi}_2, \hat{\phi}_3) \tag{2}$$

We fuse the feature vectors with the neighborhoods to reduce the resolution while retaining information. *avg* is the average pooling with kernel size $k$ and striding $s$, as shown in Equation (3).

$$\hat{\phi} = avg(\hat{\phi}_{\{1:3\}}, k, s) \tag{3}$$

Each $c$-dimensional feature slice at positions $i \in \{1, ..., h'\}$ and $j \in \{1, ..., w'\}$ is the patch feature vector of the corresponding image-patch at the spatial position. $c$, $h'$, $w'$ are the channel, height, width of $\hat{\phi}$.

### 3.2. Trimming Patch-Level Gaussian Distribution

For all samples in the training set, we obtain the patch feature vectors $x_{i,j}$ at patch position $(i,j) \in [1, h'] \times [1, w']$, and the set of feature vectors $X_{i,j} = \{x_{i,j}^n | n \in [1, N]\}$, $N$ is the number of samples in the training set. We can estimate a Gaussian distribution with parameters $\mu_{i,j}$ and $\Sigma_{i,j}$ based on $X_{i,j}$, where $\mu_{i,j}$ is the sample mean in $X_{i,j}$ and $\Sigma_{i,j}$ is the covariance matrix shown in Equations (4) and (5). In order to ensure that the covariance matrix is invertible for the subsequent Mahalanobis distance calculation, we added the term $\lambda I$. $I$ is an identity matrix, $\lambda = 1 \times 10^{-6}$.

$$\mu_{i,j} = \frac{1}{N} \sum_{n=1}^{N} x_{i,j}^n \tag{4}$$

$$\Sigma_{i,j} = \frac{1}{N-1} \sum_{n=1}^{N} (x_{i,j}^n - \mu_{i,j})(x_{i,j}^n - \mu_{i,j})^T + \lambda I \tag{5}$$

After estimating a Gaussian distribution at each patch position $(i, j)$, we obtain a Gaussian map. As shown in Figure 1, in the patch-level distribution, the patches containing anomalies are in the low-density region. In robust statistics, trimming methods [6] remove part of the maximum–minimum samples to obtain a statistical model with better robustness. Inspired by [15], we use the Mahalanobis distance in Equation (6) to measure the distance of a patch feature $x_{i,j}$ to the center of the Gaussian distribution as the anomaly score. A high anomaly score indicates that the patch is in a low-density region, and a low anomaly score indicates that the patch is in a high-density region. Inspired by the trimming method in robust statistics, we exclude the samples that contain patches in low-density regions.

$$M(x_{i,j}) = \sqrt{(x_{i,j} - \mu_{i,j})^T \Sigma_{i,j}^{-1} (x_{i,j} - \mu_{i,j})} \tag{6}$$

$$M_{i,j}^{norm} = \frac{M_{i,j} - M_{i,j}^{min}}{M_{i,j}^{max} - M_{i,j}^{min}} \tag{7}$$

As shown in Figure 4, sorting the obtained Mahalanobis distance from smallest to largest, a line segment with a sudden increase in slope can be found. We can calculate a distance threshold from this line segment. All patches with Mahalanobis distance greater than this threshold are considered anomaly patches.

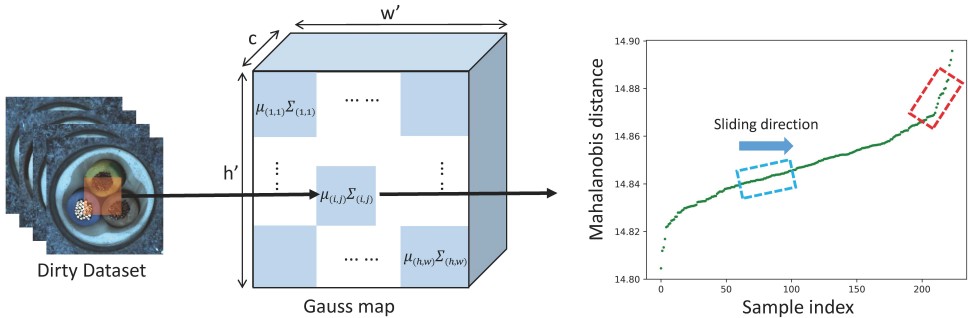

**Figure 4.** At position $(i, j)$, a Gaussian distribution is fitted with the patch feature vector of all samples. The dashed box represents the sliding window and the red dashed box represents the target window.

It is important to choose an appropriate threshold of the Mahalanobis distance to perform trimming operations. If the threshold is too low, too many normal samples will be excluded, and anomaly samples will not be effectively excluded if the threshold is too high. To make the distances on a similar scale, we use Min–Max scaling in Equation (7) for each group of Mahalanobis distances.

This paper uses the sliding-window method to implement an adaptive threshold. A window slides from left to right, and the slope of the line connecting the two endpoints of the window is considered the slope of the line segment inside the window. As the window slides to the right, we save the slope of each line segment in the window and calculate the mean value of the saved slope $Slope_{mean}$. We define the window with a slope larger than $v \times Slope_{mean}$ as the target window. The value of $v$ will be described in Section 3.3.2.

In order to find the threshold of the Mahalanobis distance more accurately, the target window occurs $U$ times consecutively, and these target windows are saved as sequence $SEQ$. We use the mean Mahalanobis distance of all samples within the first window of $SEQ$ as the threshold.

$$L = \alpha N \tag{8}$$

$$S = \frac{L}{\beta} \tag{9}$$

$$U = \frac{L}{S} + 1 \tag{10}$$

$$T_{i,j} = \frac{1}{L} \sum_{l=1}^{L} W_{i,j}^{l} \tag{11}$$

As shown in Equations (8)–(11), $L$ is the length of the sliding window, $N$ is the number of samples in the dataset, $S$ is the sliding window step. $U$ is the number of consecutive occurrences of the target window, which can be used to ensure that the sliding window can pass the jump points in the scatter plot. $\alpha$ and $\beta$ are hyperparameters, $T_{i,j}$ is the Mahalanobis distance threshold at patch position$(i, j)$. $W_{i,j}$ is the first window in the target window sequence $SEQ$ at patch position $(i, j)$.

$$Label^n = \begin{cases} 0, & \text{if } M(x_{i,j}^n) < T_{i,j}, i \in [1, h'], j \in [1, w'] \\ 1, & \text{else} \end{cases}, n \in [1, ..., N] \tag{12}$$

$$D_{clean} = \{x^n | Label^n = 0, n \in [1, N]\} \tag{13}$$

After determining distance thresholds, we can select normal samples from the dirty dataset. In Equation (12), $Label^n$ is the label of sample $n$ in the dataset, 0 indicates normal sample, 1 indicates anomaly sample. Considering that anomaly regions have different scales, fixed-scale patch regions do not apply to all anomaly region scales. If an anomaly region is slightly larger than the patch scale, it is difficult to ensure that adjacent patches containing subtle anomalies are filtered out. It is insurance that we exclude this sample once a sample contains an anomaly patch. When the Mahalanobis distance of all patches of the sample $x^n$ is less than the corresponding threshold, we consider sample $x^n$ as a normal sample. Otherwise, we consider sample $x^n$ as an anomaly sample. According to this criterion, we can select some samples from the dirty dataset, and these selected samples are considered normal samples which constitute the clean set $D_{clean}$ shown in Equation (13).

### 3.3. Iterative Filtering

3.3.1. Iterative Processing

As shown in Figure 5, the Gaussian map is estimated based on the dirty dataset. After filtering, we can obtain a clean sample set $D_{clean}$. The new Gaussian map is estimated based on $D_{clean}$ and used to refilter the dirty dataset. The above iteration is executed until the set of clean samples $D_{clean}$ no longer changes or the number of $D_{clean}$ is more than half of the total number of the dirty dataset.

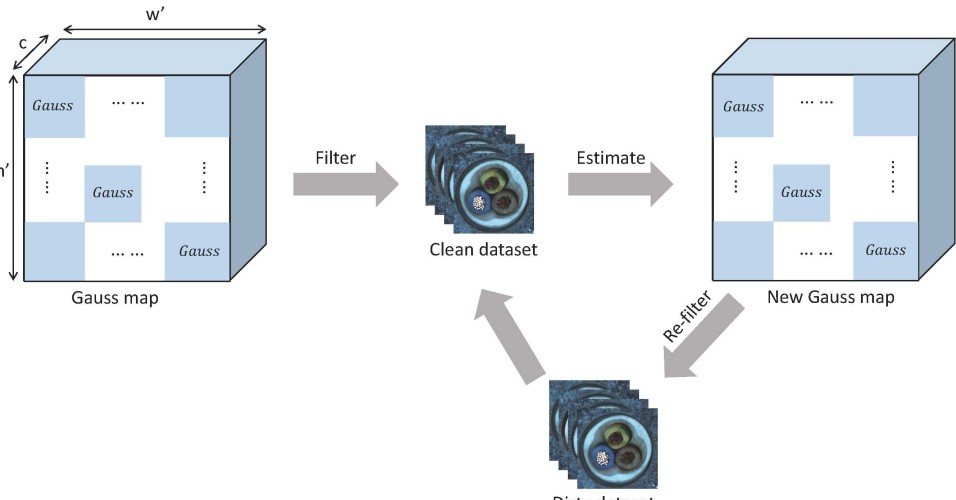

**Figure 5.** After estimating Gaussian distribution for each position on the dirty dataset, the iterative process is started.

The above iterative filtering process shown in Figure 2 can select normal sample set $D_{clean}$ from the dirty dataset. The new Gaussian map estimated based on $D_{clean}$ is used for anomaly segmentation.

### 3.3.2. Different Strategy for Sliding Window

In the first and other iterations, we used slightly different strategies to determine the slope threshold of the target window.

As mentioned before, we define the window with a slope greater than a multiplier of the historical mean of the slope as the target window. The value of this multiplier directly affects the determination of the target window. In order to determine this multiplier reasonably, we need to know the trend of the slope in the scatter plot. As shown in Figure 6, the scatter plot at the first iteration is not the same as the scatter plot at a certain iteration. For the first iteration, the starting point is the point with the index at $N \times 1/3$. For a certain iteration, the starting point is the point with a small offset from the jump point. The points between the start point and the endpoint are divided into three segments. We define Equation (14) to measure the trend in the slope of the scatter plot. In Section 3.2, the parameter $v$ concerning the slope threshold $v \times Slope_{mean}$ is defined as Equation (15).

$$R = \frac{Slope_{line3}}{(Slope_{line1} + Slope_{line2}) \times \frac{1}{2}} \tag{14}$$

$$v = \frac{R - \tau}{\eta} + \tau \tag{15}$$

In Equation (14), we use the ratio of the slope of line3 to the mean of the slopes of line1 and line2 to judge the slope trend. If the slope ratio $R$ is close to 1, the line between the starting point and endpoint can be considered a straight line, and we can directly choose the endpoint as the Mahalanobis distance threshold. If the slope ratio $R$ is much greater than 1, it means that the slope of the line between the starting point and the endpoint grows large. It indicates the existence of some patches far away from the Gaussian distribution. In Equation (15), $\tau$ is the hyperparameter. If the slope ratio $R$ is less than $\tau$, the slope ratio $R$ can be considered close to 1. Otherwise, a value between $\tau$ and $R$ is taken as the slope threshold $v$ to search for the target window. The hyperparameter $\eta$ is greater than 1, ensuring that $v$ is between $\tau$ and $R$. We tested the effect of different values of $\eta$ on the performance in Section 5.2.

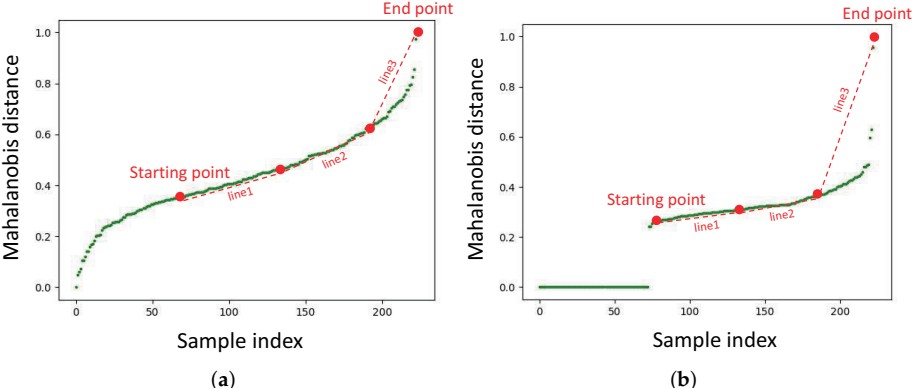

(**a**)                                                        (**b**)

**Figure 6.** (**a**) The sorted Mahalanobis distance plot for a certain patch in the first iteration. (**b**) The sorted Mahalanobis distance plot of a certain patch at a certain iteration.

### 3.4. Anomaly Detection and Segmentation

For a test sample $x_{test}$, the Mahalanobis distance between its patch feature vector $x_{i,j}$ at position $(i, j)$ and the center of the Gaussian distribution at the position $(i, j)$ is used as the anomaly score of the image patch corresponding to the position $(i, j)$. A two-dimensional matrix anomaly map is obtained.

We select the maximum value of the anomaly map as the image-level anomaly score. The anomaly score of each pixel is obtained by restoring the anomaly map to the size of the original image using bilinear interpolation, as shown in Figure 7.

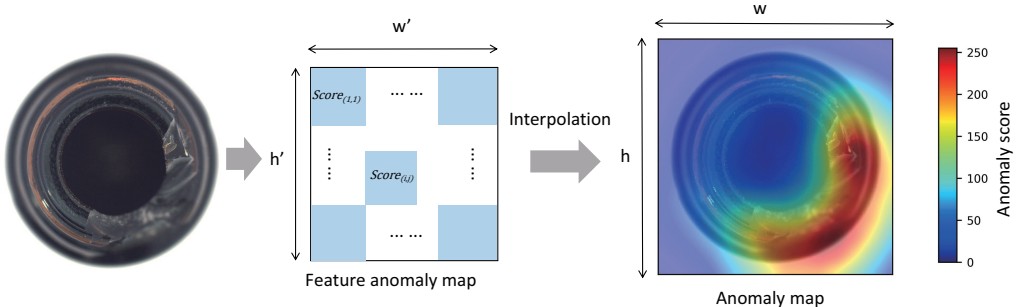

**Figure 7.** Using Mahalanobis distance as the anomaly score yields the feature anomaly map. The anomaly map is obtained by upsampling the feature anomaly map.

### 3.5. Flowchart of PaIRE

In Figure 8, all training samples are samples in the dirty dataset. The patch-level Gaussian distributions are estimated based on all samples. For the first iteration, we use the sliding window for the first iteration to compute the Mahalanobis distance threshold. We can obtain clean samples from the dirty dataset based on the Mahalanobis distance threshold. We re-estimate the patch-level Gaussian distribution based on clean samples, and we use the sliding window for other iterations to compute the Mahalanobis distance threshold. After that, we can select clean samples from the dirty dataset. The above iterative process stops when the number of selected clean samples is unchanged, or the number of clean samples is more than half of the number of samples in the dirty dataset. The patch-level Gaussian distributions estimated based on the final clean samples are used for anomaly detection.

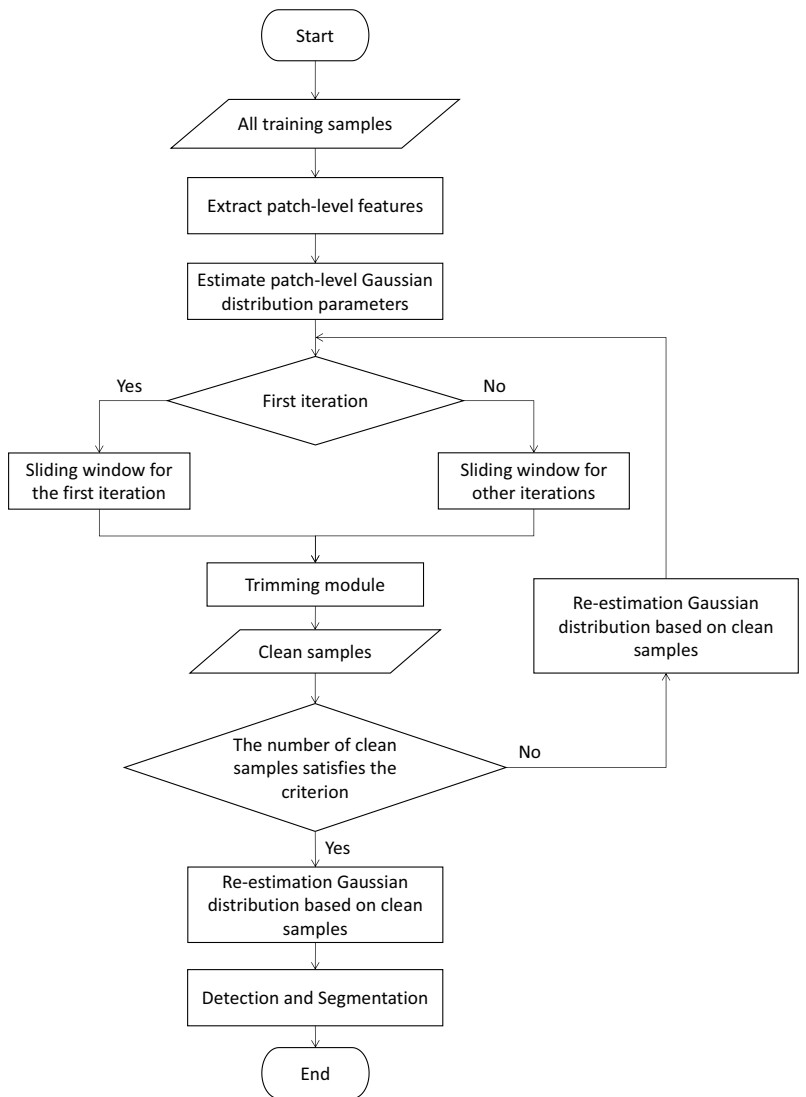

**Figure 8.** Flowchart of PaIRE.

## 4. Experiment

This part is divided into three sections: construction of the dirty datasets, comparison experiments, and discussion. Each section will be described below.

### 4.1. Construction of the Dirty Datasets

In order to construct dirty datasets, we modified the three datasets. These datasets are MVTec, BTAD, and KolektorSDD2. The way to modify each dataset into a dirty dataset will be described below.

- MVTec [35]: An industrial anomaly detection dataset containing 10 object categories and 5 texture categories. Each category has a test set with multiple real anomaly samples and a training set with only normal samples.
  To simulate the dirty dataset, we randomly selected a certain number of anomaly samples from the test set to be added to the training set and removed an equal number of normal samples from the training set. The test set remained unchanged. The number of randomly selected anomaly samples occupy 5%, 10%, 15%, 20%, 25%, and 30% ratio of the training set. We use the data augmentation by rotation and flip if there are insufficient anomaly samples in the test set. The details of the composition of the dirty MVTec dataset are shown in Table 1.

- BTAD [36]: BeanTech Anomaly Detection Dataset. The dataset contains a total of 2830 real-world images of three industrial products, which showcased body and surface defects with the size of 600 × 600. The training set only contains normal samples. The same approach simulates the dirty dataset as MVTec was used. If the real anomaly samples are insufficient, the same rotation and flip data augmentation are used.
- KolektorSDD2 [37]: A surface-defect detection dataset with over 3000 images containing several types of defects obtained while addressing a real-world industrial problem. In the original KSDD2 dataset, the training set contains 10% of the anomaly samples. On the KSDD2 dataset, we use the same strategy as before to simulate the dirty dataset with noise ratios of 15%, 20%, 25%, and 30%.

**Table 1.** Components of the dirty MVTec dataset. The upper row is the number of normal samples and the lower row is the number of anomaly samples. The bolded font means that the number of the anomaly samples in the test set were insufficient and we used data augmentation for them.

| Category | 5% | 10% | 15% | 20% | 25% | 30% |
|---|---|---|---|---|---|---|
| bottle | 198<br>11 | 188<br>21 | 178<br>31 | 167<br>42 | 157<br>52 | 146<br>63 |
| cable | 213<br>11 | 201<br>23 | 190<br>34 | 179<br>45 | 168<br>56 | 157<br>67 |
| capsule | 208<br>11 | 197<br>122 | 186<br>33 | 175<br>44 | 164<br>55 | 153<br>66 |
| carpet | 266<br>14 | 252<br>28 | 238<br>42 | 224<br>56 | 210<br>70 | 196<br>84 |
| grid | 250<br>14 | 238<br>26 | 224<br>40 | 211<br>53 | 198<br>**66** | 185<br>**79** |
| hazelnut | 371<br>20 | 352<br>39 | 332<br>59 | 313<br>**78** | 293<br>**98** | 274<br>**117** |
| leather | 233<br>12 | 221<br>24 | 208<br>37 | 196<br>49 | 184<br>61 | 172<br>73 |
| metal nut | 209<br>11 | 198<br>22 | 187<br>33 | 176<br>44 | 165<br>55 | 154<br>66 |
| pill | 254<br>13 | 240<br>27 | 227<br>40 | 214<br>53 | 200<br>67 | 187<br>80 |
| screw | 304<br>16 | 288<br>32 | 272<br>48 | 256<br>64 | 240<br>80 | 224<br>96 |
| tile | 219<br>11 | 207<br>23 | 196<br>34 | 184<br>46 | 173<br>57 | 161<br>69 |
| toothbrush | 57<br>3 | 54<br>6 | 51<br>9 | 48<br>12 | 45<br>15 | 42<br>18 |
| transistor | 202<br>11 | 192<br>21 | 181<br>32 | 170<br>**43** | 160<br>**53** | 149<br>**64** |
| wood | 235<br>12 | 222<br>25 | 210<br>37 | 198<br>49 | 185<br>**62** | 173<br>**74** |
| zipper | 228<br>12 | 216<br>24 | 204<br>36 | 192<br>48 | 180<br>60 | 168<br>72 |

*4.2. Comparison Experiments*

4.2.1. Experiments on Dirty MVTec Dataset

We compared our method with three others: CutPaste, PaDim, DFR. The image was resized to $256 \times 256$ for training and inference. In formula (3), we set $k = 10$, $s = 4$. In Equations (8) and (9), we set $\alpha = 0.05$, $\beta = 3$. In Equation (15), we set $\tau = 1.15$ and $\eta = 2.3$.

We tested the Image-level detection performance of the compared methods at noise ratios of 5%, 10%, 15%, 20%, 25%, and 30%, respectively. As shown in Figure 9, the y-axis is the mean image-level AUROC value for all categories, and the x-axis is the different noise ratios. Our proposed PaIRE method outperforms all comparison methods when the dataset contains noise. PaDim and CutPaste are sensitive to noise and suffered from severe performance degradation. Detailed comparison data with different noise ratios are shown in Table 2. At each noise ratio, the bold format indicates the best performance.

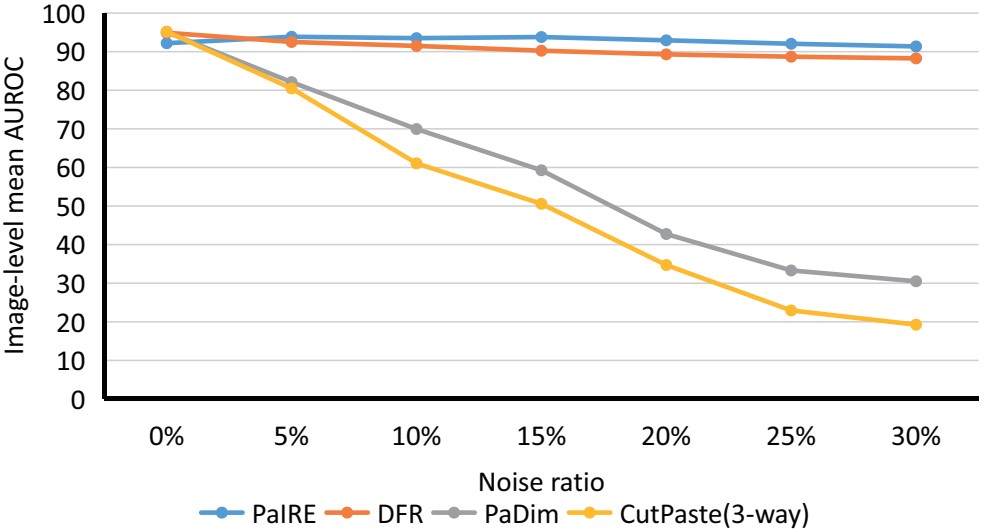

**Figure 9.** Comparison on MVTec dataset.

**Table 2.** Comparison with SOTA methods at different noise ratios.

| Noise Ratio | PaIRE | PaDim | DFR | CutPaste |
|---|---|---|---|---|
| 0% | 92.19 | 95.06 | 94.90 | **95.2** |
| 5% | **93.86** | 82.11 | 92.51 | 80.48 |
| 10% | **93.50** | 69.95 | 91.49 | 61.06 |
| 15% | **93.79** | 59.27 | 90.25 | 50.58 |
| 20% | **92.94** | 42.75 | 89.30 | 34.71 |
| 25% | **92.05** | 33.31 | 88.70 | 22.94 |
| 30% | **91.36** | 30.5 | 88.26 | 19.26 |

The anomaly segmentation performance of PaIRE at different noise ratios is shown in Figure 10. As shown in Table 2 and Figure 10a, it can be seen that both image-level detection and pixel-level detection have only a slight performance degradation when the noise ratio is larger than 15%. As shown in Figure 10b, PaIRE can precisely locate the anomaly area in the image.

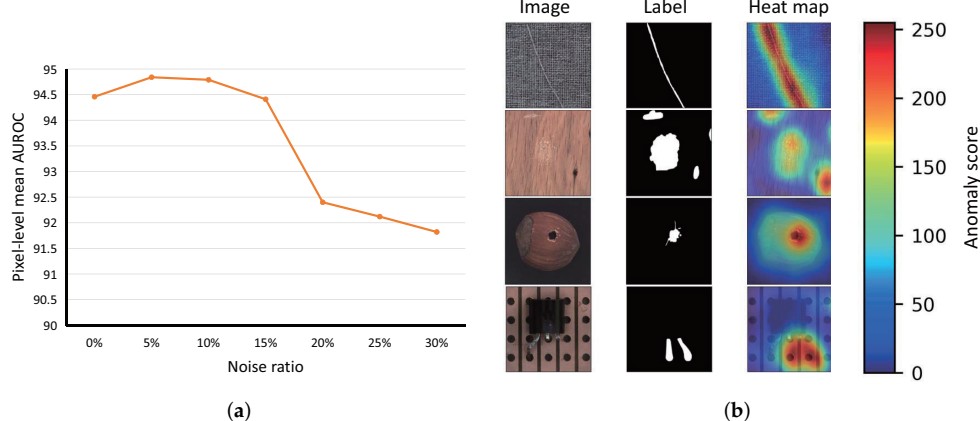

(**a**)                                                    (**b**)

**Figure 10.** (**a**) Pixel-level mean AUROC at different noise ratios. (**b**) Segmentation result at a noise ratio of 15%.

### 4.2.2. Experiments on Dirty BTAD Dataset

For the comparison on the BTAD dataset, all hyperparameters are the same as for the MVTec dataset. The image is resized to $256 \times 256$ for training and inference. The comparison results are shown in Figure 11.

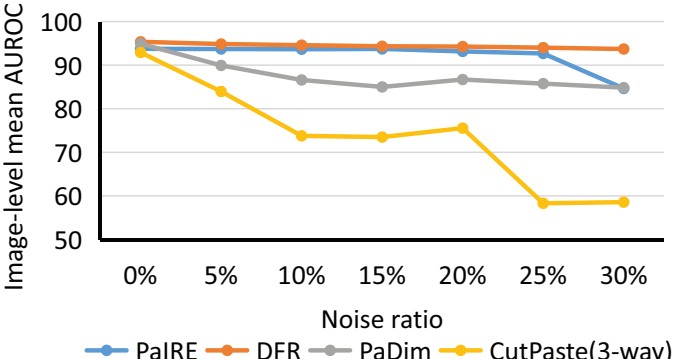

**Figure 11.** Comparison on BTAD dataset.

As shown in Figure 11, PaIRE achieved state-of-the-art results except at a 30% noise ratio. Both PaIRE and DFR show good noise resistance on the BTAD dataset. PaDim and CutPaste are heavily affected by noise, the same as for the performance on the MVTec dataset.

### 4.2.3. Experiments on Dirty KSDD2 Dataset

For the comparison on the KSDD2 dataset, we used the same hyperparameters as before, except for the image size. The image is resized to $224 \times 224$ for training and inference.

The original KSDD2 dataset contains 10% anomaly samples, so we started our comparison from a 10% noise ratio. As shown in Figure 12, PaIRE also achieved a state-of-the-art performance on the KSDD2 dataset. The hyperparameters of PaIRE based on MVTec tuning achieved a state-of-the-art performance on both BTAD and KSDD2 datasets, indicating the generality of the hyperparameters.

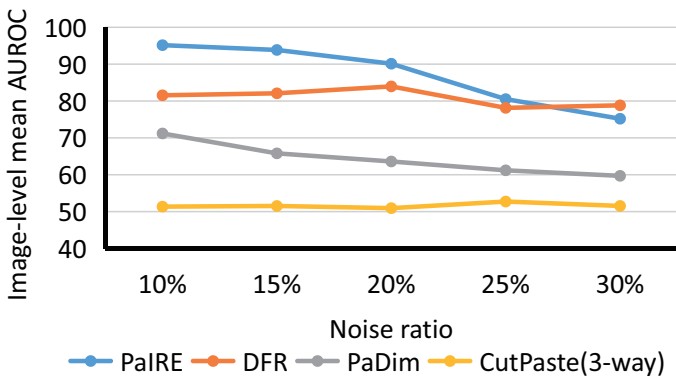

**Figure 12.** Comparison on KSDD2 dataset.

### 4.3. Discussion

We will discuss the results of the comparison experiments on MVTec, BTAD, and KSDD2 datasets.

PaDim is a parametric probabilistic model, and it needs to estimate the Gaussian distribution of normal samples. However, PaDim does not consider the noisy samples in the dataset when estimating the parameters of the Gaussian distribution, and the estimated Gaussian distribution deviates severely from the distribution of normal samples, resulting in a degradation of the model performance.

The self-supervised method CutPaste is sensitive to noisy samples because it considers all samples in the dataset as normal samples, creates artificial anomaly samples by applying special data augmentation to the samples in the dataset, and trains a classifier to discriminate normal samples from artificial anomaly samples. When the dataset contains noise, the noisy samples will generate false labels to mislead the classifier resulting in a degradation of the model performance.

The reconstruction method DFR uses a pretrained network to extract the feature map and reconstruct it using an autoencoder. DFR uses PCA to determine the size of the latent space of the autoencoder. Benefiting from the noise reduction effect of PCA, DFR has better noise resistance than PaDim and CutPaste.

Our proposed PaIRE belongs to the probabilistic model, which can estimate the Gaussian distribution of normal samples from the dirty dataset by filtering the noisy samples and increasing the number of clean samples. PaIRE achieves advanced results in all three datasets.

## 5. Ablation Study

### 5.1. Trimming Method and Iterative Filtering

Our proposed framework PaIRE has two modules for anomaly samples in the training set: trimming method in robust statistics and iterative filtering. We add the trimming module and the iterative filtering module one by one to explore the contribution of these modules. We gradually tested on MVTec to explore the impact of the module on the average AUROC of all categories at all noisy ratios.

As shown in Figure 13, When no module is added, the anomaly samples severely affect the model performance. After adding the trimming module, the model performance is improved tremendously. This shows that the trimming module can effectively exclude noisy samples so that the estimated Gaussian distribution is closer to the distribution of normal samples. After adding the iteration module, more clean samples are selected so that the estimated distribution is further closer to the distribution of normal samples.

As shown in Figure 14, The x-axis indicates the different noise ratios, and the y-axis indicates the mean of the number of clean samples of all categories. After adding the iteration module, more clean samples were selected from the dirty dataset, but the model performance increased only slightly, as shown in Figure 13. This is because most of the normal samples are similar, and more similar samples do not increase the model performance significantly.

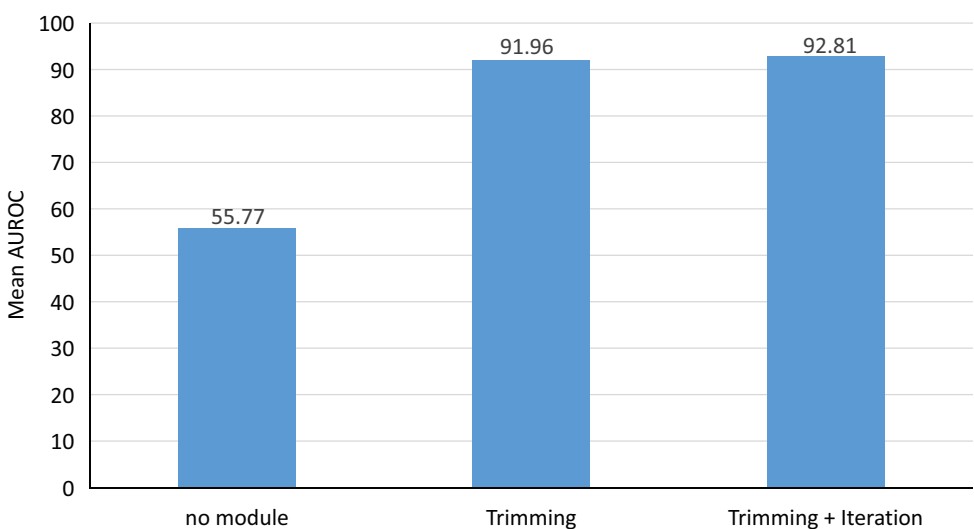

**Figure 13.** Trimming and iteration module performance testing.

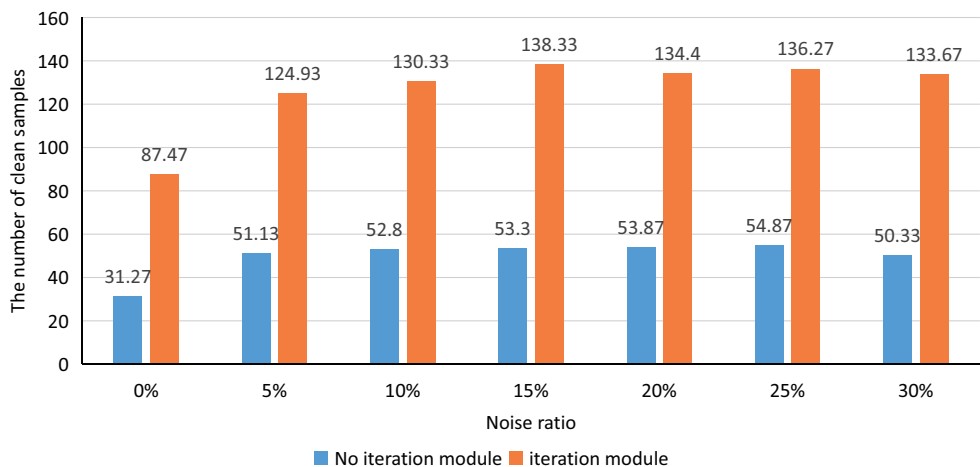

**Figure 14.** Comparison of the number of clean samples with and without the iteration module.

### 5.2. Effect of Mahalanobis Distance Threshold on Performance

We performed experiments to investigate the effect of hyperparameter $\eta$ in Equation (15), which relates to the slope threshold $v * Slope_{mean}$. The $v * Slope_{mean}$ is used to determine the target window. With $k = 10$, $s = 4$ in Equation (3), we tested the performance of image-level detection for different $\eta$. The experiments are based on the MVTec dataset.

As shown in Figure 15, The x-axis represents the different hyperparameter $\eta$, and the y-axis represents the mean value of image-level AUROC at 5%, 15%, and 25% noise ratio. In Equation (15), when $\eta$ is small, the Mahalanobis distance threshold is large. It leads to anomaly samples not being effectively excluded, and the estimated Gaussian distribution deviates from the distribution of normal samples. When $\eta$ is large, the Mahalanobis distance threshold is small, leading to a small number of selected clean samples, and the estimated Gaussian distribution cannot accurately match the distribution of normal samples. The ablation experiment shows that the average performance is highest when $\eta = 2.3$.

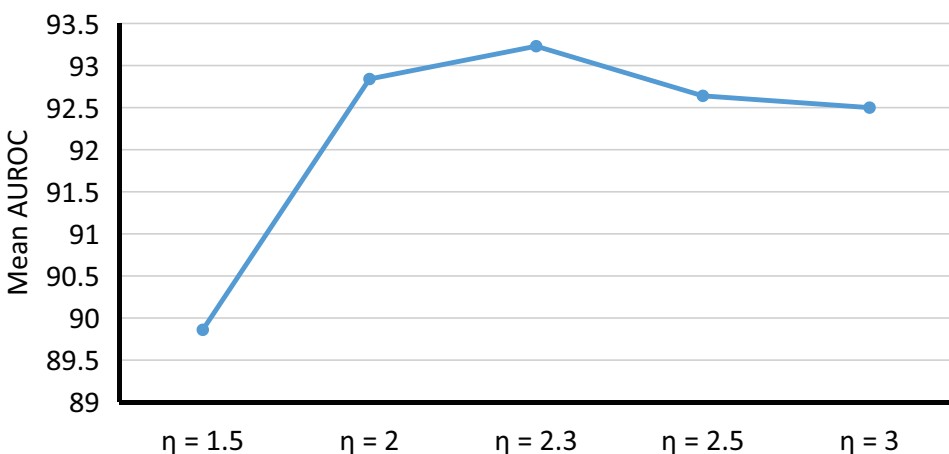

**Figure 15.** Effect of hyperparameter $\eta$ on performance.

## 6. Conclusions

We propose an unsupervised anomaly detection and segmentation framework, PaIRE, in the case of training datasets containing noise samples. PaIRE reduces the effect of noisy samples on the estimation of Gaussian distribution parameters by filtering them from the dirty dataset. Compared to the state-of-the-art methods, PaIRE has better noise resistance and anomaly-detection performance.

The methods based on image patches have a limitation. These methods assume that patches at the same position should be independently identically distributed. The texture category and aligned category satisfy this assumption, but the nonalignment category may not satisfy it. Aligning targets in image space or feature space to improve anomaly-detection performance will be our next work.

**Author Contributions:** Supervision, L.W.; methodology, J.G.; data preparation, X.Y.; writing—original draft, J.G.; writing—review, L.W. All authors have read and agreed to the published version of the manuscript.

**Funding:** This research received no external funding.

**Data Availability Statement:** All datasets used in this paper are publicly available, MVTec at https://www.mvtec.com/company/research/datasets/mvtec-ad/ (accessed on 10 May 2021), BTAD at http://avires.dimi.uniud.it/papers/btad/btad.zip (accessed on 3 November 2021), KSDD2 at https://www.vicos.si/resources/kolektorsdd2/ (accessed on 9 November 2021).

**Acknowledgments:** No external contributions or funding.

**Conflicts of Interest:** The authors declare no conflict of interest.

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
