# Peer review of "Unsupervised Anomaly Detection and Segmentation on Dirty Datasets"

_futureinternet, doi:10.3390/fi14030086_

Round 1
Reviewer 1 Report
The article is already a re-submitted version of the manuscript that I have reviewed before. Therefore, I appreciate the efforts of the authors to improve the quality of the manuscript. Indeed, the paper has changed a lot – in the good direction. The presentation is now more clear. I am satisfied with done revisions.
In my opinion, the article is almost suitable for publication.
Some minor comments include:
Line 52: please rephrase the sentence (twice “proposed”).
Fig. 5: I cannot see what is written inside the cubes.
Line 237: replace “5” with “five”.
Author Response
Response to Reviewer 1 Comments
Point 1: Line 52: please rephrase the sentence (twice “proposed”).
Response 1: I revised the sentence. The word "proposed method" was changed to "model".
Point 2: Fig. 5: I cannot see what is written inside the cubes.
Response 2: Figure 5 has been replaced with a larger size image. For the sake of clarity, I denote the Gaussian distribution parameter μ, Σ in the cubes as "Gauss".
Point 3: Line 237: replace “5” with “five”.
Response 2: "10" and "5" have been replaced with "ten" and "five".
Thanks for your suggestions!

Reviewer 2 Report
I was expecting from the authors a response letter with point-to-point comments addressing the reviewer's questions
Author Response
Response to Reviewer 2 Comments
Sorry for the missing response letter.
I was encouraged to revise the manuscript extensively and resubmit it. I packaged the revised manuscript with a response letter and resubmitted it. It is possible that the response letter was not uploaded because the review system can only upload one pdf file at the review stage.
I apologize for the missing response letter. I will add the content of the previous response letter below.
Suggestion 1.
In the introduction, please state the contributions of this study and clearly state the gap in the literature.
Response 1.
We have added relevant content in the introduction.
Suggestion 2.
What is the main advantage of the proposed approach? Please give some explanations about it?
Response 2.
The experimental section demonstrates that the proposed method has good noise resistance and anomaly detection performance, and the reasons for this are explained in an added discussion section.
Suggestion 3.
Line 38. Please add/discuss references to corroborate the following statement. "In practice, most of the existing unsupervised methods have severe performance degradation when the training set contains anomaly samples."
Response 3.
References were added to line 34 of the revised manuscript.
Suggestion 4.
Please check the equations labels and formatting.
Response 4.
We modified the expression of some formulas and checked the formula labels and formatting.
Suggestion 5.
Please add a flowchart of the computational model proposed in this manuscript to improve the paper's readability.
Response 5.
We have added flowcharts and described the whole process in Section 3.5.
Suggestion 6.
Line 220:Please define normal samples in the context of this paper?
Response 6.
We define the normal samples in lines 21-22 of the manuscript.
Suggestion 7.
Line 290: the term "tremendously" sounds exaggerated here. Please discuss why the trimming module is effective and improves the robustness of the model.
Response 7.
We explain why the trimming module is effective in Manuscript 323-327.
Suggestion 8.
The results and discussion section are written clearly. However, the authors should show the limitations of the proposed approach.
Response 8.
We explain the limitations of the proposed method in the conclusion part.
Suggestion 9.
The experiments in section 5.2 only describe what is observed in the figure and provide no further discussion.
Response 9.
We have added a related explanation in Manuscript 341-346.
Suggestion 10.
The conclusion is weak and vague based on the experiments presented in this paper. It does not reflect what was developed in the study and can be improved, highlighting the contributions of the manuscript.
Response 10.
We modified the conclusion part.
